# Room-temperature photonic logical qubits via second-order nonlinearities

Stefan Krastanov [1,2✉], Mikkel Heuck[1], Jeffrey H. Shapiro [1], Prineha Narang [2], Dirk R. Englund [1] &
Kurt Jacobs[3,4,5]

Recent progress in nonlinear optical materials and microresonators has brought quantum computing with bulk optical nonlinearities into the realm of possibility. This platform is of great interest, not only because photonics is an obvious choice for quantum networks, but also as a promising route to quantum information processing at room temperature. We propose an approach for reprogrammable room-temperature photonic quantum logic that significantly simplifies the realization of various quantum circuits, and in particular, of error correction. The key element is the programmable photonic multi-mode resonator that implements reprogrammable bosonic quantum logic gates, while using only the bulk $\chi^{(2)}$ nonlinear susceptibility. We theoretically demonstrate that just two of these elements suffice for a complete, compact error-correction circuit on a bosonic code, without the need for measurement or feed-forward control. Encoding and logical operations on the code are also easily achieved with these reprogrammable quantum photonic processors. An extrapolation of current progress in nonlinear optical materials and photonic circuits indicates that such circuitry should be achievable within the next decade.

[1] Department of Electrical Engineering and Computer Science, Massachusetts Institute of Technology, Cambridge, MA 02139, USA. [2] John A. Paulson School of Engineering and Applied Sciences, Harvard University, Cambridge, MA 02138, USA. [3] U.S. Army Research Laboratory, Computational and Information Sciences Directorate, Adelphi, MD 20783, USA. [4] Department of Physics, University of Massachusetts at Boston, Boston, MA 02125, USA. [5] Hearne Institute for Theoretical Physics, Louisiana State University, Baton Rouge, LA 70803, USA. ✉email: stefankr@mit.edu

Any attempt to build coherent quantum hardware is met with the relentless deleterious influence of the environment. To combat it, all of today's nascent quantum computers must be cooled to cryogenic temperatures. Superconducting quantum circuits require dilution refrigerators to eliminate thermal noise[1,2], and ion trap processors are cooled to <10 K to reduce collisions with stray gas molecules[3]. This need for cooling poses a problem for many potential applications of quantum information processing; it greatly reduces the prospects for portable devices, and significantly impacts the cost and practicality of large-scale deployment as repeaters and routers for communication networks. Even optical circuits that employ single-site defects (e.g., color centers or rare-earth impurities) require cryogenic temperatures to reduce thermal line broadening[4–6]. So too do linear optics schemes that employ detectors as their sole nonlinear element (in this case to avoid the overhead incurred by inefficient detection)[7,8].

At present, there are only a few platforms that appear to have the potential for quantum processing at both room temperature and pressure[9–12]. We explore photonic circuits that employ bulk optical nonlinearities as their nonlinear element is a particularly promising one. Bulk nonlinear elements not only do not suffer from thermal excitation, but due to their size they are less affected by thermal broadening. Until recently, the possibility of realizing quantum devices with bulk nonlinearities seemed remote, due both to the weakness of these nonlinearities and the problem of wave-packet distortion[13–18]. Substantial progress in the effective strength of the material nonlinearities, the introduction of ultra-confining cavities[19–21], and a relatively simple solution to wave-packet distortions[22–24] have changed that outlook.

Achieving the physical technology to implement nonlinear photonic quantum circuits is not the only challenge to realizing room-temperature quantum logic. For practicality, one must implement this logic using the strongest available nonlinearity, the leading-order $\chi^{(2)}$ nonlinear susceptibility, and for efficient room-temperature operation the logic and error correction circuits should avoid measurements or feed-forward control. Two basic approaches to information processing with photons are possible. The first is the use of single- or dual-rail encoding in which each mode contains no more than one photon[25]. While this has the advantage that all circuit constructions from the well-developed qubit model can be employed, this leads to complex circuits even for correcting the loss of a single photon. The smallest code for this purpose uses five modes (ten for dual-rail encoding)[26,27]. While there is little work on minimal circuits for correcting the five-qubit code, from circuits for the seven-qubit Steane code, we estimate that it requires a least nine additional modes and >30 CNOT gates. The alternative is to use bosonic codes that employ multiple photons per mode, but in this case it is far from obvious what gates and circuits are required to implement the error correction, let alone how to realize these gates with a $\chi^{(2)}$ interaction. While explicit error correction procedures for bosonic codes have been elucidated[28–32], they all involve non-demolition or photon-number-resolving measurements. It is not yet known how to construct the unitary multiphoton operations required to replace such measurement using only a $\chi^{(2)}$ nonlinearity, or the complexity of doing so. The only unitary circuit that has been explicitly constructed to date to correct a bosonic code is in the form of a 40-layer neural network using an idealized $\chi^{(3)}$ medium[33].

Here, we propose an approach for implementing all-unitary, and thus room-temperature, quantum logic on multimode multiphoton states using only a fixed $\chi^{(2)}$ nonlinearity. This paradigm, which employs as its basic module a single triply-resonant cavity with a time-dependent drive, significantly reduces the complexity of the physical circuits required to implement

multiphoton quantum logic in general, and error correction in particular. The joint operation performed on the three modes by the module is controlled by the time-dependent drive. In this way, the module is able to perform a wide range of three-mode multiphoton gates. We demonstrate the power of this approach by explicitly constructing a measurement-free error-correcting circuit for a two-mode bosonic code. This circuit requires just two of our three-mode modules, along with some controllable linear elements. Our compact unitary circuits do not employ any measurements or feed-forward control, which makes them particularly useful for fast quantum routers and repeaters. However, measurements will certainly be required to read out a message or the results of a computation. Fortunately, it is straightforward to use unitary circuits in general, and our processor in particular, to enable high-fidelity measurements at room temperature, even when only inefficient detectors are available. To do so, it is enough to use a unitary circuit to map a single photon to a sufficiently large number of photons that can then be detected. This amplification can be implemented rapidly using a doubling process. First a $\chi^{(2)}$ nonlinearity is used to convert one photon in one mode 1 to two photons in a second mode via down-conversion. Second, a frequency conversion process (which employs a $\chi^{(2)}$ and a classical pump) is then used to transfer each of the two photons back to the first mode. Repeating this photon-doubling cycle provides exponentially fast amplification. Since measurements are only to be used at the end of a computation, the additional overhead for amplification remains small. Thus, while we do not analyze this measurement method in detail here, it is clear that the lack of efficient photon detectors is not an obstacle to room-temperature quantum information processing.

In the next section, we describe the control Hamiltonian realized by the driven triply-resonant cavity that forms our basic processing module and give examples of important gates that can be implemented by the module. Then we show how a full error correction process can be built from a small number of these multiphoton gates. Lastly, we discuss the materials science and fabrication challenges that must be addressed, in order to realize our loss-correction circuit. By extrapolating the rate of progress in these areas over the past decade, we estimate a timeline for demonstrating this circuit.

## Results

**A controllable three-mode cavity**. We consider three resonant modes of a cavity in a $\chi^{(2)}$ medium, with respective mode operators $\hat{a}$, $\hat{b}$, and $\hat{c}$, and frequencies $\omega_a$, $\omega_b$, and $\omega_c$. We neglect any dissipative dynamics until later sections, where we discuss hardware implementations. The cavity is driven by a coherent classical pump with frequency $\omega_p$. We depict it in Fig. 1, in which the pump may be a microwave frequency electric field or an optical drive. By choosing the frequencies to satisfy

$$\omega_a = 2\omega_b, \tag{1}$$

$$\omega_c + \omega_p = \omega_b, \tag{2}$$

the $\chi^{(2)}$ medium couples the modes via the Hamiltonian

$$\hat{H}_{nl}(t) = \hbar\left[\chi\hat{a}\hat{b}^{\dagger 2} + g(t)\hat{b}^{\dagger}\hat{c} + \text{H.c.}\right], \tag{3}$$

in which $\chi$ is the coupling rate of the $\chi^{(2)}$ nonlinearity, $g(t)$ is the coherent amplitude of the classical pump, and we have moved to the rotating frame of the oscillators. Since Schrödinger's equation contains $H/\hbar$, it is the rates $\chi$ and $g(t)$ that determine the dynamics. If we measure time in units of $1/\chi$ then all rates are divided by $\chi$, and the dynamics is determined by

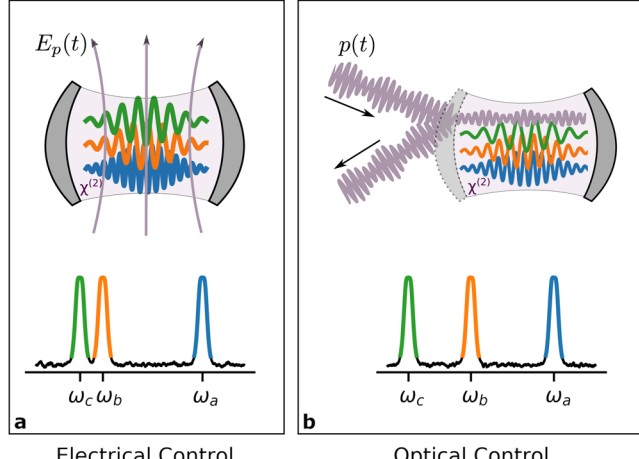

**Fig. 1 The triply-resonant nonlinear cavity.** The $\chi^{(2)}$ medium enables the joint control of three modes. We denote the mode operators respectively by $\hat{a}$, $\hat{b}$, and $\hat{c}$. The $\chi^{(2)}$ medium enables frequency doubling from $\hat{b}$ to $\hat{a}$, and a three-way interaction between modes $\hat{b}$, $\hat{c}$, and the control field. The control field is either **a** a classical microwave drive, $E_p(t)$, or **b** a classical optical drive of envelope $p(t)$. This three-way interaction is effectively a linear interaction between modes $\hat{b}$ and $\hat{c}$ that is controlled by the classical drive. The combination of the fixed frequency-doubling interaction and the controlled linear interaction allows extensive control of the joint nonlinear evolution. This evolution conserves the quantity $2n_a + n_b + n_c$, in which $n_a$, $n_b$, and $n_c$ are the occupation numbers of the respective modes. We also depict the relative values of the frequencies of the three modes; in **a**, the frequencies of modes $\hat{b}$ and $\hat{c}$ are separated by the much smaller frequency of the microwave drive.

$$\frac{\hat{H}_{\mathrm{nl}}(t)}{\hbar\chi} = \hat{a}\hat{b}^{\dagger 2} + p(t)\hat{b}^{\dagger}\hat{c} + \mathrm{H.c.}, \quad (4)$$

in which we have defined $p(t) = g(t)/\chi$. Thus up to a scaling of time, the dynamics is entirely determined by the rate parameter $p(t)$. We will not have to introduce any additional rate parameters (and thus any additional timescales) until we consider loss in the section on hardware considerations. In that section, we will express our rate parameters in terms of the physical parameters of realistic devices.

Note that the second term in $\hat{H}_{\mathrm{nl}}(t)$, which is controlled via the amplitude of the pump, is merely a linear coupling between modes $\hat{b}$ and $\hat{c}$. This interaction cannot by itself generate a universal set of quantum gates[34,35]. It turns out, however, that it can do so when combined with the time-independent frequency-doubling interaction.

We denote the number of photons in the three modes respectively by $n_a$, $n_b$, and $n_c$, and the corresponding operators for the photon number by $\hat{n}_a$, $\hat{n}_b$, and $\hat{n}_c$. Since the Hamiltonian commutes with $2\hat{n}_a + \hat{n}_b + \hat{n}_c$, the value of that observable is preserved. The Hamiltonian cannot, therefore, mix subspaces defined by different integer values of $2n_a + n_b + n_c$. Nevertheless, it does provide complete control within each subspace by virtue of the fact that the repeated commutators of $\hat{a}\hat{b}^{\dagger 2}$ and $\hat{b}^{\dagger}\hat{c}$ generate a complete Lie algebra for all such subspaces[28,36–38]. It is this fact that provides the power of our processing unit.

In general, to implement quantum gates between the three modes, we will need to generate a set of distinct evolutions, one for each of the $2\hat{n}_a + \hat{n}_b + \hat{n}_c = \mathrm{const}$ subspaces. We can do that with a single control pulse, as for each subspace, there are many choices for $p(t)$ that generate the same unitary operation. We can

use numerical search methods to find a control function $p(t)$ that simultaneously generates the required evolution for each of the set of subspaces. Naturally, we wish to find the control that implements a given gate in the shortest time, a challenge solved as described below.

Lastly, the modes of this quantum processor will need to be actively coupled to the waveguides that carry the quantum states to be processed. Otherwise, the process of capturing the content of the waveguides will be too slow due to the necessarily high quality factor of the modes of the processor. To actively couple the cavity modes to the waveguides, we envision using the method given in Heuck et al.[23].

**Compiling unitary operations**. To find the control pulse $p(t)$ required to implement a given unitary operation, we employ numerical search methods, an approach often referred to as optimal control[39–41]. We introduce a parameterization for $p(t)$ as a piecewise-constant signal in which the duration of each interval is variable. This parameterization is essential because the always-on frequency-doubling component of the Hamiltonian necessitates optimizing the length of the pulse. In order to avoid unphysical pulses, we constrain both the duration and amplitude of each interval by the use of sigmoid functions. The full expression for the resulting unitary operation is

$$\hat{U}(\mathbf{v}) = \prod_{l=1}^{s} \exp\left\{-i\left[\mathrm{f}(X_l, P_l)\hat{b}^{\dagger}\hat{c} + \sigma(T_l)\hat{a}\hat{b}^{\dagger 2} + \mathrm{H.c.}\right]\right\}, \quad (5)$$

where

$$\mathrm{f}(X_l, P_l) = \arctan(X_l) + i\arctan(P_l), \quad (6)$$

$$\sigma(T_l) = \frac{\Delta\tau}{1 + \exp(-T_l)}, \quad (7)$$

and $\mathbf{v} = \{X_l, P_l, T_l: l = 1, \ldots, N\}$ is the set of parameters that defines the pulse. The parameters $\{X_l : X_l \in \mathbb{R}\}$ and $\{P_l : P_l \in \mathbb{R}\}$ are related to the quadrature of the pulse, which is constrained to the interval $[-1, 1]$ by arctan, while the $\{T_l : T_l \in \mathbb{R}\}$ are related to the duration of each segment, which is constrained to the interval $[0, \Delta\tau]$. We fix the number of piecewise-constant intervals, $s$, as well as the relative unitless time scale $\Delta\tau$.

Consistently good performance is obtained even with $s < 60$. This permits the use of standard automatic differentiation tools, without the need for approximations, such as GRAPE[40]. Our parameterization also has the advantage that it does not allow for pathological pulses. Once we have obtained a piecewise constant control function for a given gate, we use GRAPE and standard regularization techniques to smooth out the pulse, ensuring it has both reasonable bandwidth and power. Throughout the optimization, the robustness of the control to calibration errors is verified. The time scale $\Delta\tau$ is shortened until a threshhold is reached at which the control pulse is no longer robust. The above approach to generating control functions, together with a number of symbolic optimizations, will be presented in detail in a related tutorial[39].

**Examples of programmable gates**. Through the use of our implicitly constrained optimal control method, we can perform with high fidelity any gate that keeps $2n_a + n_b + n_c$ constant. If the length of the control pulse is unconstrained, and dissipation is neglected, we can achieve fidelities arbitrary close to unity. For gates reported here, we constrain the duration of the control pulses as much as possible before reaching unitary fidelities <0.999. In later sections considering hardware implementations, we also describe the effects of dissipation. Here, we describe a number of important unitary operations that fulfill that

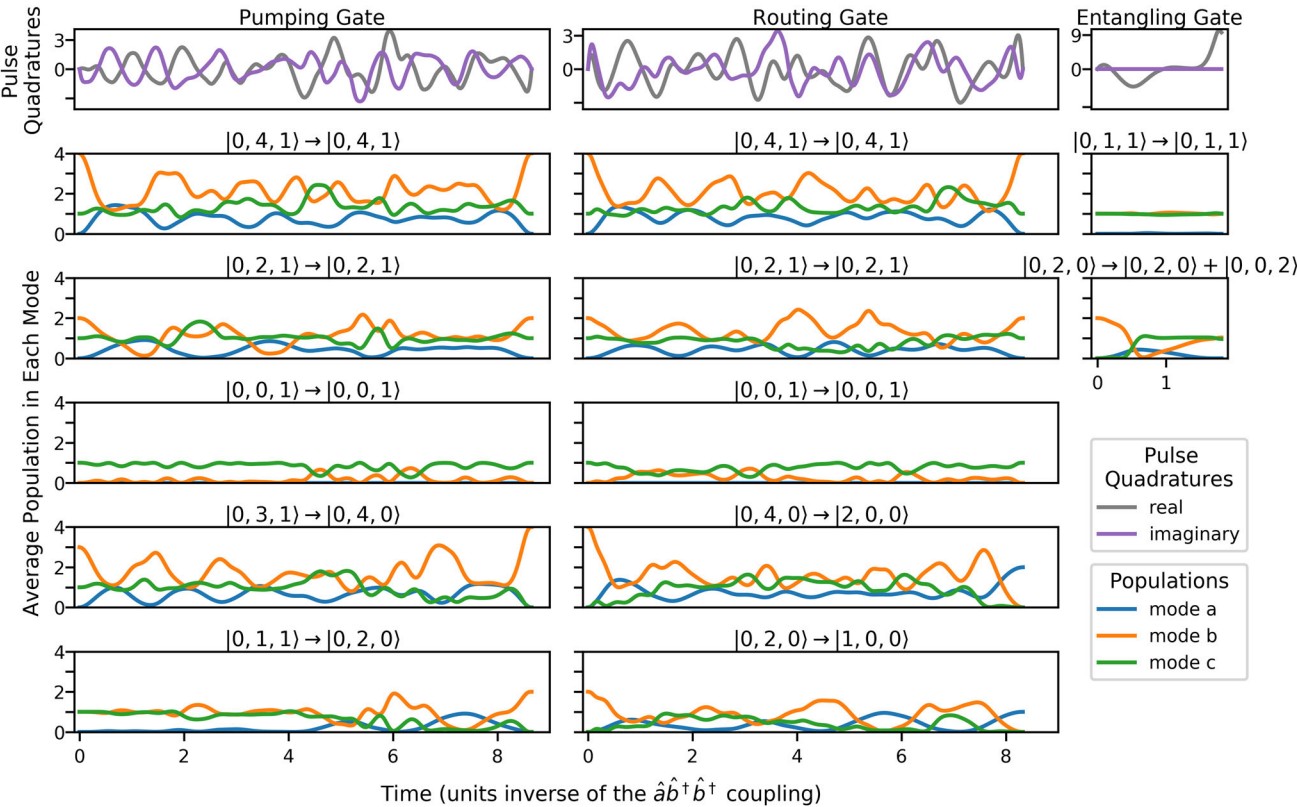

**Fig. 2 The control pulses implementing the three gates that are used to build our error correction circuit.** The top row shows the real and imaginary parts of the control pulses for each gate. The following rows show how the populations of the modes evolve under each gate for a given initial state. The optimizer produces pulses $p(t)$ such that each of the desired transformations leads to constructive interference at the exact same time. Shorter pulses are possible, at the expense of higher power and bandwidth requirements[39], up to a point at which the pulse is too short to perform even a single complete oscillation in a subspace defined by an integer value of $2n_a + n_b + n_c$.

constraint, some of which are also depicted in Fig. 2. More general unitary operations can be performed by reshuffling the modes of the three-mode processors, as seen in later sections. Given the long cavity lifetimes requires for these operations, reshuffling necessitates rapid catch, and release of photons from and into the connected waveguides, e.g., by using active control as done in Heuck et al.[23].

Throughout the following paragraphs, we will use the notation $|n_a n_b n_c\rangle$ to denote a Fock state with $n_a$, $n_b$, and $n_c$ photons in modes $\hat{a}$, $\hat{b}$, and $\hat{c}$, respectively.

We begin with the Toffoli Gate, which is a three-qubit non-Clifford gate, distinguished by the fact that together with just the single-qubit Hadamard gate it enables universal quantum circuits[42,43]. Of particular relevance for our purposes is the fact that it usually requires six two-qubit CNOT gates to implement[44,45], while our realization requires only a single application of the three-mode processor. We realize the gate in the Hadamard basis (i.e., our gate is a phase gate with two control qubits) for photonic qubits encoded in a single- or dual-rail configuration. In this basis, the Toffoli unitary maps all joint Fock states to themselves except for the state $|111\rangle$ to which it applies a $\pi$ phase.

We also define a conditional routing gate as one that swaps the state of two modes depending on the state of a third mode. This class of gates is useful for breaking down conditional multi-qudit operations into smaller units. We first route the target mode to a particular waveguide, based on the state of the control mode, and we perform the appropriate single-mode quantum operation in the new physical location of the target mode. Such routing is indispensable, if our goal is to avoid measurements in error-

correcting circuits, as measurements usually require hardware at cryogenic temperatures. Typically, a non-demolition measurement is performed by entangling the required information with an ancilla, and performing a demolition measurement on the ancilla. The result, a classical bit, is then fed forward through a classical computer that decides what quantum operation to perform next. We avoid the measurement and classical decisions through coherent quantum feedback[46,47], where we simply perform a multimode quantum gate conditioned on the ancilla. The realization for the routing gate suggested below is what we use in our bosonic error-correcting circuit, but other setups are feasible as well. Below $|n_a n_b n_c\rangle$ denotes a Fock state with $n_a$, $n_b$, and $n_c$ photons in modes $\hat{a}$, $\hat{b}$, and $\hat{c}$, respectively. The $\hat{c}$ mode is the control, the $\hat{b}$ mode is the input, and $\hat{a}$ and $\hat{b}$ are the possible outputs:

$$|040\rangle \mapsto |200\rangle, \tag{8}$$

$$|020\rangle \mapsto |100\rangle, \tag{9}$$

$$|041\rangle \mapsto |041\rangle, \tag{10}$$

$$|021\rangle \mapsto |021\rangle, \tag{11}$$

$$|001\rangle \mapsto |001\rangle. \tag{12}$$

When used in the error-correcting circuits described in later sections, mode $\hat{c}$ will contain an ancillary photon on which routing will be conditioned, while mode $\hat{b}$ will contain one of the

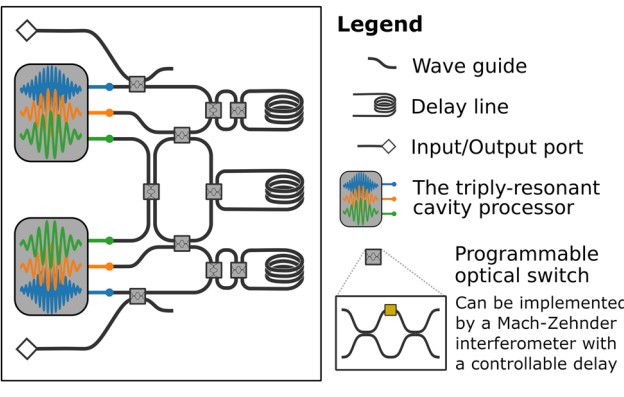

**Legend**

⌇ Wave guide

▱ Delay line

◇ Input/Output port

The triply-resonant cavity processor

▣ Programmable optical switch

Can be implemented by a Mach–Zehnder interferometer with a controllable delay

**Fig. 3 Our minimal architecture for error correction of bosonic codes, readily expandable to larger tasks.** The circuit depicted can be used to correct a single-photon loss using a two-mode bosonic code. The circuit consists of two cavity processors, which for the most part process each mode of the code separately, and a small network of reprogrammable beam splitters and delay lines. These are used to reroute states between the modes of processors as necessary. Each cavity processor is also capable of performing many multi-qubit gates for single- and dual-rail encoded qubits, as well as preparing and manipulating higher-number Fock states. The network of programmable beam splitters between the processors and the delay lines can also be expanded to a fully connected network, enabling universal rerouting between the three modes of each processor for general-purpose quantum computation. The programmable beam splitters can be implemented as Mach–Zehnder interferometers (as shown in the inset) with two 50/50 beam splitters and a programmable delay (the orange medium in the diagram). Classical electronics will be necessary to ensure the pacing of various operations in this device, but no feedback or decision circuitry is necessary, as the approach is measurement free.

two modes of our multimode bosonic code. A second processor will be used for other modes.

Focusing further on the error-correcting functionality, we need a gate that can correct for photon loss in a codeword. For the code we employ, we require the gate to preserve the states $|001\rangle$, $|021\rangle$, and $|041\rangle$, and accomplish the mapping

$$|031\rangle \mapsto |040\rangle, \quad (13)$$

$$|011\rangle \mapsto |020\rangle. \quad (14)$$

This operation is necessary for reverting photon loss in the code mode stored in $\hat{b}$, while storing information about the occurrence of that loss in mode $\hat{c}$. Again, we will need two three-mode processors, each acting on one of the modes making up our error-correcting code. Each of the physical modes of the code will be stored in the corresponding $\hat{b}$ oscillators.

To complete our error correction circuit, we use a gate that entangles two modes. We require this operation because one of the code words is an entangled state, and the loss of a photon breaks this entanglement. This gate provides the mapping

$$|011\rangle \mapsto |011\rangle, \quad (15)$$

$$|020\rangle \mapsto \frac{|020\rangle + |002\rangle}{\sqrt{2}}. \quad (16)$$

This gate is also a symmetrizing operation for the state of the modes $\hat{b}$ and $\hat{c}$. It is these two oscillators $\hat{b}$ and $\hat{c}$ that will contain the two modes of our error-correcting code.

The above gates are only a few of the many operations that the triply-resonant cavity processor can perform. Among these gates are those important for the processing of unprotected single-photon states, and operations that enable unitary modification

and number-resolved measurements on modes with more than one photons, including bosonic codes. Importantly, these operations are performed with a single use of the triply-resonant cavity, while otherwise they would require complete circuits with multiple discrete operations. This leads to drastically simpler overall circuits, at the expense of requiring this more sophisticated and difficult to fabricate triply-resonant optical resonator.

**Measurement-free error correction.** We demonstrate the versatility of our control protocol by constructing an error-correcting circuit around the three-mode processor. The circuit we obtain is not only simple and short, but it also does not require any measurement operations or classical feed-forward control.

We choose the following code, encoding a single qubit in two separate (spatial) bosonic modes, whose logical states are given by

$$|1\rangle_{\rm L} = |22\rangle, \quad (17)$$

$$|0\rangle_{\rm L} = \frac{|40\rangle + |04\rangle}{\sqrt{2}}. \quad (18)$$

This two-mode code allows correction for the loss of a single photon from either mode. For a channel that has a 10% probability of a single-photon loss for each mode this implies an 81% chance of transmission without error, 18% chance of transmission with a correctable error, and a 1% chance of transmission with an uncorrectable error. We choose this code because it is possible to perform the correction process with operations that conserve the quantity $2n_{\rm a} + n_{\rm b} + n_{\rm c}$, so long as one is judicious in choosing these operations.

We must first consider the effect of a photon loss on the code. The loss of a photon on the first mode is described by the action of $\hat{a}\hat{I}$. This transforms the initial code state $|C\rangle = \alpha|0\rangle_{\rm L} + \beta|1\rangle_{\rm L}$ into the error state $|E_1\rangle = \alpha|12\rangle + \beta|30\rangle$. Similarly, the loss of a photon from the second mode produces the error state $|E_2\rangle = \alpha|21\rangle + \beta|03\rangle$. For each of these two errors, we need to perform a different correction procedure. Typically, this is achieved by a non-demolition measurement that projects the state of the system onto either the logical subspace or one of the error subspaces, followed by a unitary correction operation conditioned on the measurement result. We sidestep these requirements by using coherent control. We employ two quantum ancillas, initialized to contain single photons, on which routing gates will be conditioned. Thus, our correction procedure involves the following steps. First, we put the information about the presence of an error in the ancillas by using two conditional pumping gates acting in parallel (the code modes are each placed in a $\hat{b}$ mode, while the ancillary photons are in the corresponding $\hat{c}$ modes), resulting in the following transformation of the overall ancillas-code state:

$$|11\rangle \otimes |C\rangle \mapsto |11\rangle \otimes |C\rangle, \quad (19)$$

$$|11\rangle \otimes |E_1\rangle \mapsto |01\rangle \otimes |F_1\rangle, \quad (20)$$

$$|11\rangle \otimes |E_2\rangle \mapsto |10\rangle \otimes |F_2\rangle, \quad (21)$$

where $|F_1\rangle = \alpha|22\rangle + \beta|40\rangle$ and $|F_2\rangle = \alpha|22\rangle + \beta|04\rangle$. The feed-forward solution would have measured the ancillas and performed different operations depending on the measurement, but as already mentioned that would be slow and require additional cooled hardware and classical decision circuitry. Instead, we perform the following unitary operation (as before, the left multiplier in the tensor product $|c_1 c_2\rangle$ denotes the content of the ancillary $\hat{c}$ modes of each of the two processors, and $|C\rangle$

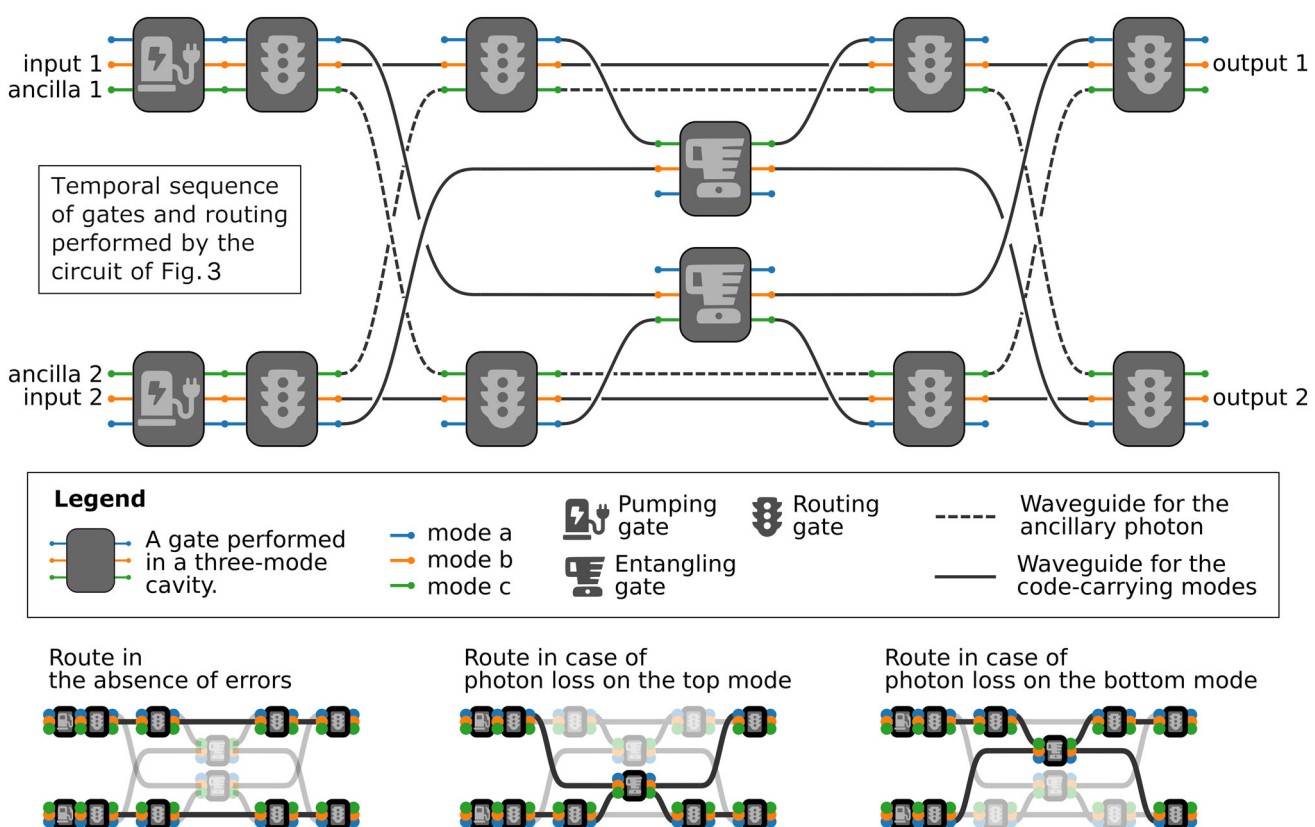

**Fig. 4 The error-correcting circuit, unrolled in time.** The horizontal axis represents the flow of time, depicting how a pair of triply-resonant cavities is being used. This circuit would be executed on the hardware depicted in Fig. 3. The main drawing is the sequence of operations that we need to perform in parallel in the two triply-resonant cavities, in order to perform the error correction. After placing the code and ancilla modes in the appropriate cavity modes, we accomplish the initial pumping and routing gates. After that, we need to shuffle the ancillary modes by releasing them in the appropriate waveguides. The spatial modes into which the code states are moved depend on the state of the ancillas, thanks to the conditional routing gates. As the ancillas contain information about the presence of photon-loss errors, this lets us perform operations conditioned on the loss of a photon, by performing the two conditional branches in parallel in different physical locations of the circuit. The conditional routing gates then act in reverse, ensuring that all spatial modes end in the same location, without breaking the bijectivity required for any quantum circuit. The various spatial modes employed can be seen in the bottom insets of the figure. Supplementary Figure 1 provides a more detailed rendition. Importantly, as seen in Fig. 3, we do not need 12 triply-resonant cavities as depicted above, rather only 2 cavities with a network of waveguides and programmable beam splitters[72] that can route the spatial modes as necessary, so that each cavity can be used repeatedly. The gate pictographs are taken from the Font Awesome icon set.

denotes the two modes of the error-correcting code, stored in the $\hat{b}$ oscillators of the two processor):

$$|11\rangle \otimes |C\rangle \mapsto |11\rangle \otimes |C\rangle, \qquad (22)$$

$$|01\rangle \otimes |F_1\rangle \mapsto |01\rangle \otimes |C\rangle, \qquad (23)$$

$$|10\rangle \otimes |F_2\rangle \mapsto |10\rangle \otimes |C\rangle. \qquad (24)$$

Without the ancillas, this operation would be impossible as it would break the bijectivity of the unitary operator by mapping many states to one. The conditional routing gates are crucial for the performance of this operation—depending on the ancillas, they route the modes containing the code to different spatial modes that perform $|F_1\rangle \mapsto |C\rangle$ and $|F_2\rangle \mapsto |C\rangle$ independently and in parallel. The conditional routing gates then ensure that all three paths end up in the same spatial modes at the end of the circuit. The error-correcting circuit can be seen in Fig. 3 as a suggested physical layout, and in Fig. 4 as a sequence of abstract gates.

**Encoding operation**. Encoding a qubit in the two-mode code is particularly simple using the three-mode processor. To do so we

have to perform the operation

$$|00\rangle \mapsto |22\rangle, \qquad (25)$$

$$|10\rangle \mapsto \frac{|40\rangle + |04\rangle}{\sqrt{2}}. \qquad (26)$$

Given that we already have access to the entangling gate, encoding can be done by putting the unprotected photonic qubit in cavity $\hat{c}$ and putting ancilla photons in cavities $\hat{a}$ and $\hat{b}$. Then, we perform the partial encoding gate

$$|111\rangle \mapsto |200\rangle \qquad (27)$$

$$|110\rangle \mapsto |110\rangle, \qquad (28)$$

thus mapping the state $\alpha|1\rangle + \beta|0\rangle$ in $\hat{c}$ to the precursor of the two-mode code $\alpha|11\rangle + \beta|20\rangle$ in $\hat{a}$ and $\hat{b}$. As before, here $|n_a n_b n_c\rangle$ denotes a Fock state with $n_a$, $n_b$, and $n_c$ photons the modes $\hat{a}$, $\hat{b}$, and $\hat{c}$ of a single processor. Turning this into the complete code state requires a simple application of the entangling operation already discussed above. These two operations preserve the constant of motion $2\hat{n}_a + \hat{n}_b + \hat{n}_c$ and as such can

be compiled to a single control pulse performed in a single triply-resonant cavity.

**Two-qubit logical operations.** Single-qubit rotations in the logical space of the two-mode code can be realized by using our three-mode processors, as such rotations do preserve $2n_a + n_b + n_c$. Moreover, two-qubit logical operations can also be performed. For instance, consider a CPHASE gate, which together with the single-qubit rotations form a universal set. We need to perform the operation

$$|2\underline{2}\rangle \otimes |\underline{2}2\rangle \mapsto -|2\underline{2}\rangle \otimes |\underline{2}2\rangle, \tag{29}$$

while mapping all other combinations of $\left\{|2\underline{2}\rangle, \frac{|4\underline{0}\rangle + |0\underline{4}\rangle}{\sqrt{2}}\right\}^{\otimes 2}$ to themselves. Our three-mode processor can perform this operation by acting on just two of the four modes making up the two logical qubits. If we index each mode as, e.g., $|2_i 2_j\rangle \otimes |2_k 2_l\rangle$, we need to act only on modes $j$ and $k$, by first transferring them to modes $b$ and $c$ of the cavity, and designing a pulse to perform the operation

$$|022\rangle \mapsto -|022\rangle, \tag{30}$$

while mapping all of $|000\rangle, |002\rangle, |020\rangle, |004\rangle, |040\rangle, |024\rangle, |042\rangle$, and $|044\rangle$ to themselves. The overall CPHASE operation on the four physical modes forming the two logical qubits takes the form,

$$|2\underline{2}\rangle \otimes |\underline{2}2\rangle \mapsto -|2\underline{2}\rangle \otimes |\underline{2}2\rangle, \tag{31}$$

$$\frac{|4\underline{0}\rangle + |0\underline{4}\rangle}{\sqrt{2}} \otimes |\underline{2}2\rangle \mapsto \frac{|4\underline{0}\rangle + |0\underline{4}\rangle}{\sqrt{2}} \otimes |\underline{2}2\rangle, \tag{32}$$

$$|2\underline{2}\rangle \otimes \frac{|\underline{4}0\rangle + |\underline{0}4\rangle}{\sqrt{2}} \mapsto |2\underline{2}\rangle \otimes \frac{|\underline{4}0\rangle + |\underline{0}4\rangle}{\sqrt{2}}, \tag{33}$$

$$\frac{|4\underline{0}\rangle + |0\underline{4}\rangle}{\sqrt{2}} \otimes \frac{|\underline{4}0\rangle + |\underline{0}4\rangle}{\sqrt{2}} \mapsto \frac{|4\underline{0}\rangle + |0\underline{4}\rangle}{\sqrt{2}} \otimes \frac{|\underline{4}0\rangle + |\underline{0}4\rangle}{\sqrt{2}}, \tag{34}$$

where the underlined modes are the ones that are manipulated inside of a three-mode processor. Notice that a phase is gained only in the first row, where both of the modes in the processor (the underlined modes) have two photons. Such an operation can be performed directly by our processors or, if shorter and simpler control pulses are desired, by first using the $\chi^{(2)}$ interaction to upconvert them to lower photon numbers.

**Comparison with other approaches.** Comparisons with other codes and types of hardware require care because the various systems have significant differences. Nevertheless, we elucidate how our control protocol substantially reduces the depth of a typical circuit and removes the need for entire classes of expensive operations. As discussed in the introduction, error correction procedures have been proposed for bosonic codes, but these require non-demolition or photon-number-resolving measurements, and it has not yet been described how such measurements can be replaced by unitary operations generated by a $\chi^{(2)}$ nonlinearity. We can however, compare our circuit to the explicit correction circuit presented in Steinbrecher et al.[33].

One way to compare the efficiency of circuits is to examine how long each takes relative to the characteristic unit of time for the given hardware. The circuit we have constructed above requires six gates, for a total of 40 units of time (relative to the $\chi^{(2)}$ coupling strength), and four transfers in and out of cavities. The correction circuit employing the quantum optical neural network (QONN) architecture[33], which is the closest analog of our

hardware, requires 40 layers, resulting also in 40 units of time, but since it uses a $\chi^{(3)}$ rather than a $\chi^{(2)}$ medium, the nonlinearity is significantly weaker, so that the circuit takes longer in real time. Furthermore, the QONN circuit requires 40 transfers in and out of the nonlinear cavities (one for each layer), ten times more than our architecture.

One can instead implement photonic quantum logic by using only the vacuum and one-photon Fock states to encode qubits (i.e., a single- or dual-rail encoding). The smallest error-correcting code in this setting requires five physical qubits[26]. The logic required to determine the error syndrome for this code requires 16 CNOT gates and 4 auxiliary qubits[27]. The auxiliary qubits can either be measured, in which case the error can be determined using a classical computer, or a unitary circuit could process the auxiliary qubits and perform the correction[48–52]. For each of the 16 different values of the four-bit syndrome, a unitary correction circuit would need to perform a different correction operation. This requires quite a large number of ancillas and CNOT gates, as discussed in the introduction. Our room-temperature design thus represents a dramatic reduction in circuit size and duration. We also emphasize that using all-unitary processes, which is the approach we take here, provides a practical advantage; doing so avoids the need to introduce additional amplification and classical feedback circuitry.

Competing with "active" gate-based approaches to measurement-fee error correction, is the use of continuous autonomous QEC[53–57]. In that family of protocols, one needs to design an exotic dissipator, usually through reservoir engineering, which provides an irreversible evolution from the error-space back the code space.

**Hardware prospects.** We will introduce a less abstract model of our triply-resonant cavity design, in order to better describe the materials science and fabrication challenges it faces. This model also lets us give physical values for the unitless durations we have found above for our control pulses. We will start by describing the physical realization for the $\hat{a}\hat{b}^{\dagger 2}$ and $p(t)\hat{b}^{\dagger}\hat{c}$ terms in the Hamiltonian. Naturally, these terms requires the presence of eigenmodes $\hat{a}$, $\hat{b}$, and $\hat{c}$. The corresponding field operators would be (e.g., for the $\hat{a}$ mode)

$$\hat{\mathbf{B}}_a(\mathbf{r}) = \sqrt{\frac{\hbar\omega_a}{2}}\hat{a}\mathbf{b}_a(\mathbf{r}) + \text{H.c.}, \tag{35}$$

$$\hat{\mathbf{D}}_a(\mathbf{r}) = \sqrt{\frac{\hbar\omega_a}{2}}\hat{a}\mathbf{d}_a(\mathbf{r}) + \text{H.c.}, \tag{36}$$

where we used the magnetic field and the electric displacement in order to keep the quantization consistent in the nonlinear regime[58–60]. The $\mathbf{b}(\mathbf{r})$ and $\mathbf{d}(\mathbf{r})$ eigenmodes can be computed from classical electromagnetism and are normalized to $\int \mu_0^{-1}|\mathbf{b}|^2 d\mathbf{r} = 1$ and $\int \varepsilon_0^{-1}n^{-2}|\mathbf{d}|^2 d\mathbf{r} = 1$. The overall Hamiltonian of the system will be

$$\hat{H} = \int d\mathbf{r}\left(\frac{\hat{\mathbf{B}}^2}{2\mu_o} + \frac{\hat{\mathbf{D}}^2}{2\varepsilon_0 n^2} - \frac{\chi^{(2)}\hat{\mathbf{D}}^3}{3\varepsilon_0^2 n^6}\right), \tag{37}$$

where $n$ is the index of refraction (consult[60] for its complete treatment as a tensor with dispersion). The field operators are the sum of field operators for the modes $\hat{a}$, $\hat{b}$, and $\hat{c}$, as well as the field from the classical laser pulse $p(t)$. The first two terms from the Hamiltonian simply give us the harmonic oscillator terms, which we eliminate by moving to the corresponding rotating reference frames. The last term provides the nonlinear interactions in which we are interested. For simplicity, we first consider the undriven case, i.e., $p(t) = 0$. The driven case is

discussed in Supplementary Note 3. Expanding the nonlinear term and eliminating the nonresonant terms leaves us with

$$\hat{H}_{nl} = -\frac{\chi^{(2)}}{\sqrt{\varepsilon_0} n^3 \sqrt{V_{shg}}} \sqrt{\frac{\hbar^3 \omega_a \omega_b^2}{8}} \hat{a} \hat{b}^{\dagger 2} + \text{H.c.}, \qquad (38)$$

$$\frac{1}{\sqrt{V_{shg}}} = \frac{\int_{nl} d_a^i d_b^{j*} d_b^{k*} d\mathbf{r}}{\sqrt{\left( \int |\mathbf{d}_b|^2 d\mathbf{r} \right)^2 \int |\mathbf{d}_a|^2 d\mathbf{r}}}, \qquad (39)$$

where $\int_{nl}$ denotes integration only over the nonlinear medium and $i$, $j$, and $k$ denote the appropriate field components to integrate, depending on the nonlinear material being employed. Thus, $V_{shg}$ is the mode volume considered in second-harmonic generation (SHG) experiments. For simplicity, we are not acknowledging frequency and space dependencies in the refractive index $n$, and we are not specifying the components of the $\chi^{(2)}$ tensor being employed. This does not change the result we are pursuing.

The coupling rate in this nonlinear Hamiltonian imposes the units of time for the control pulses described in the previous section. This characteristic time needs to be compared to the cavity lifetimes, typically expressed through the $Q$ factor as $\tau = \frac{2Q}{\omega_a}$. This lets us introduce the following figure of merit for the characteristic number of operations before the environment destroys our quantum state

$$N = \sqrt{\frac{\hbar}{8\varepsilon_0}} \frac{\sqrt{\omega_a}}{n^3} \frac{Q\chi^{(2)}}{\sqrt{V_{shg}}}. \qquad (40)$$

Considering some recent SHG on-chip experiments (a $Q \sim 10^7$ in refs. [61,62] and a $V_{shg} \sim 800\,\mu m^3 \sim 2000\frac{\lambda^3}{n^3}$ with a 70 μm-radius micro-ring in refs. [63,64], at $\lambda_a \approx 750$ nm) in a typical nonlinear optics material like lithium niobate ($\chi^{(2)} \sim 31\frac{pm}{V}$), we obtain values $N \sim 0.03$, which is still too low for practical use. With $Q$ factors and mode overlaps in SHG experiments following a Moore's law (the recent progress is explored in Supplementary Note 4) and new designs lowering mode volumes by orders of magnitude[19–21,65], $N$—the number of elementary quantum operations within the cavity lifetime—could very well grow by orders of magnitude and reach tens to hundreds over the next decade. $N$ is related to a typical figure of merit in SHG experiments—the conversion efficiency[66] $\eta = \frac{P_{out}}{P_{in}^2} \propto \frac{Q^3(\chi^{(2)})^2}{V_{shg}}$, which has seen impressive improvements in the last decade (see Supplementary Note 4). With a $Q \sim 2 \times 10^8$, which is achievable in principle[67], a mode volume of $V \sim 10^{-3}\frac{\lambda^3}{n^3}$, (see refs. [20,68] for progress in mode confinement), and $\chi^{(2)} \sim 100\frac{pm}{V}$, which is between the values for lithium niobate and gallium arsenide, we achieve $N \sim 2000$ which is enough for error correction. Moreover, new fabrication techniques for thin-film materials enable much stronger effective nonlinearities than what has otherwise been achieved on-chip. While such techniques have not been explored extensively in the optical regime, these results are an encouraging indication that similar progress may well be possible for nonlinear optical materials.

To explore how such future hardware may perform, we compare the lifetime of an encoded (protected) photonic qubit to an unprotected single-rail qubit living in the same hardware. The time scale will be set by the $Q$ factor of the cavities under consideration; however, in order to present physical values for the parameters we will set $Q \sim 2 \times 10^8$ at $\lambda_a \sim 750$ nm, which is well within the thermorefractive theoretical limit[67]. In Fig. 5, we compare the performance of our error-correcting protocol to that of an unprotected single-rail qubit, and see that the error-correcting threshold is $N \sim 2000$, a very demanding value which we are

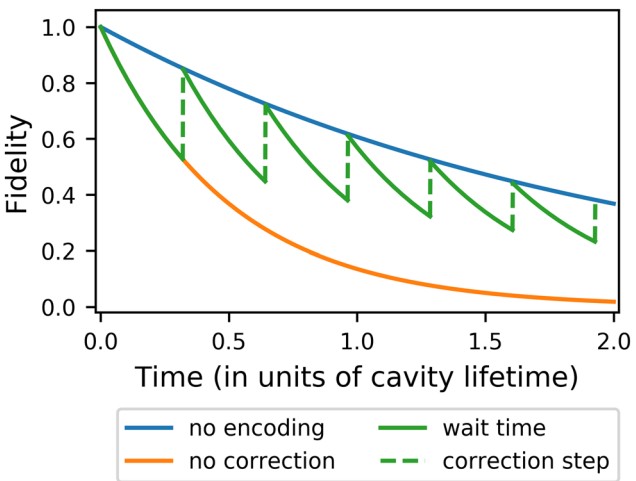

**Fig. 5 Logical qubit lifetime at the "break even" regime where it begins to outperform unprotected qubits.** In blue, we see the decay of a single photon, i.e., an unprotected single-rail qubit. In orange, we see the decay of our two-mode code if we do not perform any correction operations—it decays faster as it contains a higher number of photons. The green line represents the decay of the encoded qubit in the presence of periodic correction operations. The infidelity of the correction operations due to photon loss that can happen during the operation is taken into account. The figure represents a lower bound for the performance of our protocol, with beneficial higher order effects being neglected in order to simplify the simulation. The "break even" point is achieved at $V_{shg} \sim 10^{-3}\frac{\lambda^3}{n^3}$, $Q \sim 2 \times 10^8$, and $\chi^{(2)} \sim 100\frac{pm}{V}$ for $\lambda_a \sim 750$ nm. Waveguide losses are neglected, as they would be insignificant compared to the rest of the operations.

nonetheless optimistic about given the experimental results cited earlier. Typically for non-asymptotic codes, to achieve fault tolerance this lower-level code will have to be concatenated with an asymptotically growing stabilizer code, akin to the surface code or quantum LDPC codes and one of the many techniques for achieving non-Clifford gates (e.g., through magic states) will have to be employed. The versatility of our control protocol provides for a system agnostic to these higher-level architectural decisions.

Lastly, we need to consider the implementation of the time-dependent control pulses. In the electrical regime, the control pulse can be modulated by standard microwave electronics in CMOS-compatible hardware[69]. In the optical regime, the control pulse would have to be modulated by wave shaping through expressing the pulse in terms of its Fourier decomposition[70]. Intermediate regimes are also possible, in which we can modulate a THz electric field, by placing optically actuated Auston switches next to our triply-resonant cavities[71]. Active control will be necessary for loading and unloading photons from these long-lived cavities, e.g., by following methods proposed in Heuck et al.[23].

It is important to note that one can balance the three considerations discussed in this section: the duration, power, and bandwidth of the control pulse. When the values of all these quantities can be expressed in characteristic units close to unity, the optimization problem is well conditioned and easier to solve. Such are the control pulses we have shown (e.g., their amplitudes, bandwidths, and durations are $\lesssim 10$). However, if our hardware requires short pulses (e.g., due to low $Q$ factor), but permits high power, we can nudge the solution in this direction by reparameterizing the optimization problem[39].

## Discussion
It is accepted in the quantum computing community that any prospective purely photonic architecture for quantum information

processing would face significant challenges due to the weak photon–photon interactions available even in the best materials and resonators. Nonetheless, the present work, building upon more than a decade of theory developments on cavity-enhanced optical nonlinear interactions, shows that the monumental hardware requirements have already been nearly achieved in disparate experiments. It is an outstanding challenge to incorporate, in a single device, a record-high $Q$-factor cavity, together with extremely confined mode volumes, and fabrication-enhanced $\chi^{(2)}$ materials. However, progress over the last decade—for example, the $10^8$-fold improvement in the efficiencies of SHG—inspires confidence that this herculean task can very well be achieved within the next decade.

Moreover, our work, for the first time, shows that a single elementary photonic device can be reprogrammed on the fly to perform a set of diverse unitary operations, drastically lowering circuit complexity and depth. We have shown its applicability for typical single- and dual-rail encoded qubits, as well as its versatility in processing multiphoton Fock states. We showcased the flexibility of our control paradigm by devising an explicit error-correcting circuit for a bosonic code and the application of multi-qubit logic gates on top of that code. This is the first proposal for photonic logical qubits that includes compact encoding and correcting circuitry. Furthermore, the circuit we have designed does not require any measurement operations or feed-forward classical control, offering significant simplifications compared to a typical small stabilizer code, and opening the door for extremely fast, compact, room-temperature quantum repeaters.

## Data availability
The digitized control-pulse examples in this manuscript can be readily reproduced in most optimization toolkits (e.g., Qutip and Tensorflow under Python, or SciML and Flux under Julia). Upon request, the authors can provide these waveforms and example scripts under each of the aforementioned frameworks that produce equivalent waveforms.

## Code availability
As mentioned in the data availability statement, standard optimization toolkits were used for the creation of the control pulses and example scripts, using these frameworks can be provided upon request or found in the documentation of the aforementioned frameworks.

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

## Acknowledgements
We thank Christopher Panuski and Ryan Hamerly for many helpful conversations. Harvard Research Computing enabled much of the computational work. The SciPy, Tensorflow, and Julia open source communities provided invaluable research software. SK and MH are grateful for the funding provided by the MITRE Quantum Moonshot Programme. K.J., D.R.E., and M.H. acknowledge support from a CCDC Army Research Laboratory ECI grant.

## Author contributions
The design of the control protocol was performed jointly by the authors. S.K. wrote the optimization and analysis software, and performed the simulations. The manuscript was written by S.K. with contributions from the other authors.

## Competing interests
The authors declare no competing interests.
