## [Peer Review File · Nature Communications]

Reviewer #1 (Remarks to the Author):

The manuscript by Krastanov et al. proposed an experimental scheme to realize optical logical qubits via microcavities with strong second-harmonic nonlinearity. The authors firstly provide a method to realize a microcavity with controllable linear modal coupling and also constant second-harmonic nonlinearity. With such a system, they introduced a parameterization method for realizing arbitrary three-mode unitary operations. Especially, by the method proposed in this work, the gates for Fock states in three modes could be efficiently realized. Based on this, the authors discussed the two-mode code for logical qubit, and the measurement-free error correction scheme. At the end of this paper, the authors also discussed the trends in the research of microcavity enhanced χ^2 nonlinear interactions, and discussed the required parameters for reaching the break-even points by the two-mode code. The overall results are technically sound, and the topic appeals for broad interests from research fields including quantum optics, nonlinear optics, quantum computation and communication. Therefore, I would recommend this work publishing in Nature Communications after the authors addressed the following issues:

- (1) In Section II and III, the authors studied the system with ideal interactions without imperfections. It should be clarified at the beginning of the Sections.
- (2) It is very difficult to follow the ideas about two-mode encoding. Sometimes, the notation 000 is used, sometimes $40+04$ is used, and 22×22 also used. I could not distinguish which modes in each state. Since the authors introduce three-mode systems in their model, they should clearly illustrate which two modes are used in the encoding. Sometimes, a mode is used as ancilla, this should also be clarified.
- (3) The measurement free QEC circuit is interesting. However, a similar idea and demonstration could be found in superconducting and trapped-ion systems. The authors should discuss previous works. Especially, the autonomous QEC was demonstrated recently.
- (4) In the model presented in Fig. 3, the photons are converted between different modes and also transferred between different cavities. It seems the authors forget to discuss the input-output of the cavities. For realizing high fidelity transfer between the cavities, there should be additional controllable coupling between the cavity and bus. It should be clarified.
- (5) The references about multiple mode bosonic codes should be added.
- (6) It is not clear how to implement a gate between two logical qubits. The discussions should be extended.
- (7) For the hardware prospects, it is too long and distractive. The authors provide very detailed formula about the Hamiltonian, but they not directly related to the work presented in this work. To provide a more concrete discussion, other than hand-waving summarize of experimental progress, I would suggest the authors: First, provide the direct connection between the SHG efficiency and the N . Second, tell the guideline for experimentalists about how to optimize N . The Fig. 5&6 should be moved to supplementary materials unless the author could tell us which material is most promising and how to achieve the goal of realizing logical qubit.
- (8) A very important reference is missing: Optica Vol. 6, Issue 12, pp. 1455-1460 (2019) (a second-harmonic generation efficiency of 250,000%/W).
- (9) The results in Fig. 7 is superficial. There are several ways to improve the discussion section of this work: (i) If the system has a finite Q , the state transfer between the cavities will also be imperfect. (ii) It is still not clear how to extend the code to more modes or more cavities. Is high order code possible? (iii) Could this architecture be generalized to fault-tolerant?

Reviewer #2 (Remarks to the Author):

This manuscript is one of the most innovative and refreshing papers that I have read in the last two years. Krastanov et al. introduce a new photonic scheme for processing quantum information. The proposed advantages of this scheme are: 1) room-temperature operation; 2) realising sophisticated quantum-logic operations, such as the Toffoli gate or a conditional-routing gate, with low circuit area; and 3) likelihood of experimental realisation within the next decade. In both conception and explication this paper is a tour de force, and I expect it to have a sustained impact in quantum technology.

However there is one major claim that the paper makes repeatedly that I think the authors will have to modify. Statements such as:

“...may be the only feasible route to quantum information processing at room temperature” [Abstract] and,

“At present there appears to be only one platform with the potential for both room temperature and pressure operation: photonic circuits” [Introduction, 2nd para]

appear to be overclaims, as there appears to be other platforms with these characteristics.

Two examples are:

1. Metallic-like carbon nanospheres. Back in 2016 there was an observation of electron spin–lattice and spin–spin relaxation times of 175 ns at 300 K, published in Nature Communications (Náfrádi et al., vol. 7, 12232, 2016). This platform is now being developed commercially by Archer Materials, who in May this year signed a collaboration agreement with IBM.

2. Silicon–carbide defects. Last year there was a paper showing that an engineered quantum well can stabilize the charge state of a qubit, and compared this model to the experimentally observed room-temperature stability of the PL6 center in SiC. Again, this was published in Nature Communications (Ivády et al., vol. 10, 5607, 2019).

I had happened to read both these papers when they came out, but found them again quickly by doing a search for “room temperature quantum computer”. Of course there may be other platforms that I am unaware of, and I urge the authors to do a more comprehensive literature survey if they revise their manuscript.

With that out of the way, there are a few areas where the paper could be improved, and these arose, I think, because the authors know the material so well that they omit explanations or details that will be helpful even to expert readers. This manuscript should definitely be published, but I urge the authors to take the opportunity to clarify their presentation so as to maximise the impact of the paper.

p.1 Introduction. Second paragraph. When discussing room-temperature operation, there is no mention of the fact that currently the most efficient photon detectors are cryogenic; and

that cryogenic detectors with good number resolution are slow (transition edge sensors) or that fast cryogenic detectors have very poor number resolution (superconducting nanowire single-photon detector, SNSPD). Omitting the problem of detection at this stage in the paper appears to be a major flaw in the argument for all room-temperature operation.

p.2. Introduction. Fifth paragraph. Having made the claim that this manuscript is the only platform for room-temperature quantum information processing, the authors then effectively side-step the still open question of detection: “However, we do not explore the topic of room temperature measurements at length in this work.”

I appreciate that such an exploration would add length to an already substantial paper (and indeed could be a lengthy paper in itself) but not addressing it in at least some depth in this manuscript substantially weakens the claim of all room-temperature operation. I think that only an expert will follow the logic in the single sentence: “...a unitary circuit can be used to map the Fock state of one photon to the Fock state of N photons” and even if they do so this leaves rather large open questions regarding efficiency, speed, and circuit area of such a scheme. It may be that all the advantages of this manuscript are negated by disadvantages of room-temperature detection. Of course, that may not be the case, but there is no way to tell from the arguments here.

p.2. Methods. First paragraph. “We have set $\hbar = 1$ ”. Please do not do this. I know it is standard in theoretical quantum optics, but it is poor practice in a paper proposing a new experimental architecture, where eventually readers will wish to substitute in measured values to check the claims in the paper. (In the 90s there were several quantum optics proposals that could have been evaluated much more quickly had \hbar not been unity).

p.3. II.B.1. “We can perform with high fidelity any gate that keeps $n_a + n_b + n_c$ constant”. How high a fidelity? What limits it? In principle? In practice?

p.3. II.B.1.a. “We realise the gate in the Hadamard basis”. Where are the details of this scheme? I appreciate that you encode the three inputs of the Toffoli into the a, b, and c modes and apply your three-mode gate once, but what does $p(t)$ look like for this case? What is the fidelity, in principle? What are the effects of intra-cavity loss on the gate fidelity?

p.4. In the first 3 pages the expressions and equations are numbered, but this stops on this page. Please continue the numbering on pages 4–14.

p.4. Above the set of expressions that begins $|040\rangle \rightarrow |200\rangle$ it would be helpful if you reminded the reader of your encoding used at this point. It appears to be $|abc\rangle$, is this correct?

p.5. II.B.2. Please provide details, perhaps as a section in the Supplementary materials, to validate the claims here that this a single use of this architecture can also: 1. process unprotected single-photon states; 2. enable unitary modification of modes with high number of photons; and 3. perform number-resolved measurement on modes with high number of photons. (3. is particularly important given the earlier lack of detail on detection).

p.5. III.A. “We choose the two-mode code...” This would perhaps be clearer as: “We choose the two-spatial-mode code...”

p.5 Above the set of expressions at the bottom of the right column, that begin $|11\rangle \otimes |C\rangle$, it would be helpful if you reminded the reader of your encoding used at this point. It appears to be $|c_1 c_2\rangle \otimes |b_1 b_2\rangle$, is this correct?

pp.5–6. “The feedforward solution ... would be slow and require additional cooled hardware and classical decision circuitry”. Granted but won’t the small network shown in Figure 3 also require classical electronic control and possibly be slow? If the programmable beam splitters are implemented as Mach-Zehnder interferometers, as suggested, then the phase elements in these will have to be electronically controlled. To date such phase elements have most often be thermal, which is slow; electro-optic elements will be faster but also tend to introduce loss.

p.6 In the set of expressions at the bottom of the left column, it is not immediately obvious how are the second and third unitary operations are realised. Could the authors please provide some details?

p.7 Above the first set of expressions at the top of the left column, that begin $|00\rangle \rightarrow |22\rangle$, it would be helpful if you reminded the reader of your encoding used at this point. Is it $|b_1 b_2\rangle$?

p.7 Above the second set of expressions at the top of the left column, that begin $|111\rangle \rightarrow |200\rangle$, it would be helpful if you reminded the reader of your encoding used at this point. Is it $|abc\rangle$ again?

p.9. Hardware Prospects. Figure 7 caption and discussion in text. I’m afraid this confused me, as the Figure appears to show that encoded corrected logical qubit (green curve) at best matches the unprotected qubit (blue curve). Given the green curve never rises above the blue curve, how is the logical qubit beginning to outperform the physical qubit? This figure suggests the best thing to do is to just use the physical qubit, which I’m sure is not the intent of it.

In the text you say you “...see the error correcting threshold is $N \sim 2000$ ”. How can one see this from this figure?

p.10. Last paragraph. “...our work ... shows that a single elementary photonic device can be reprogrammed on the fly...” Is there an analysis of reprogramming time in terms of device unit times, or is reprogramming assumed to be instantaneous?

Reviewer #3 (Remarks to the Author):

The manuscript at hand theoretically describes a scheme for loss-protected optical quantum computing using bosonic codes.

The presented scheme is unique in that the error (loss) correction requires no measurements and feed-forward operations, but acts deterministically. This is enabled by presenting a novel 3-mode non-linear cavity architecture based on $X_i(2)$.

The authors first present the elements of the scheme and then describe the error-correcting circuit as well as single- and two-qubit logical operations. They also provide a survey of the existing technology and conclude that the physical parameters required for the implementation of their scheme are achievable within the next decade.

The manuscript is very well written. It is clear, presents a novel, groundbreaking scheme, and projects its future feasibility by studying the existing technologies and their development in the past few years.

I think this is an extremely good manuscript, and I recommend its immediate publication without any changes.

Well done!

Dear Editor,
Dear Reviewers,

We are grateful for the praise given to our work and doubly so for the wonderfully detailed comments and suggestions. Addressing these points made the manuscript that much stronger. Below we copy the 3 reviews (in *italic*) and address each point inline. We also inlay crops from the new manuscript itself. Changes in the manuscript are highlighted as well. The reference numbering in the final manuscript is now different.

Reviewer #1:

The manuscript by Krastanov et al. proposed an experimental scheme to realize optical logical qubits via microcavities with strong second-harmonic nonlinearity. The authors firstly provide a method to realize a microcavity with controllable linear modal coupling and also constant second-harmonic nonlinearity. With such a system, they introduced a parameterization method for realizing arbitrary three-mode unitary operations. Especially, by the method proposed in this work, the gates for Fock states in three modes could be efficiently realized. Based on this, the authors discussed the two-mode code for logical qubit, and the measurement-free error correction scheme. At the end of this paper, the authors also discussed the trends in the research of microcavity enhanced χ -2 nonlinear interactions, and discussed the required parameters for reaching the break-even points by the two-mode code. The overall results are technically sound, and the topic appeals for broad interests from research fields including quantum optics, nonlinear optics, quantum computation and communication. Therefore, I would recommend this work publishing in Nature Communications after the authors addressed the following issues:

(1) In Section II and III, the authors studied the system with ideal interactions without imperfections. It should be clarified at the beginning of the Sections.

We now explicitly acknowledge this:

II. METHODS

A. A Controllable Three-Mode Cavity

We consider three resonant modes of a cavity in a $\chi^{(2)}$ medium, with respective mode operators \hat{a} , \hat{b} , and \hat{c} , and frequencies ω_a , ω_b , and ω_c . We neglect any dissipative dynamics until later sections where we discuss hardware implementations. The cavity is driven by a coherent clas-

(2) It is very difficult to follow the ideas about two-mode encoding. Sometimes, the notation 000 is used, sometimes 40+04 is used, and 22x22 also used. I could not distinguish which modes in each state. Since the authors introduce three-mode systems in their model, they should clearly illustrate which two modes are used in the encoding. Sometimes, a mode is used as ancilla, this should also be clarified.

We have added an extra explanatory sentence to each paragraph describing a gate. However, there is no one-to-one correspondence between which mode of the code goes to which mode of the three-mode processor, as the routing of the code modes between various cavity modes is what provides much of the versatility. To explain this mapping between modes more clearly we have added a more detailed version of figure 4 to the supplementary materials.

Additions to the manuscript:

Routing gate:

$$\begin{aligned}
 |040\rangle &\mapsto |200\rangle \\
 |020\rangle &\mapsto |100\rangle \\
 |041\rangle &\mapsto |041\rangle \\
 |021\rangle &\mapsto |021\rangle \\
 |001\rangle &\mapsto |001\rangle
 \end{aligned}$$

When used in the error correcting circuits described in later sections, mode \hat{c} will contain an ancillary photon

on which routing will be conditioned, while mode \hat{b} will contain one of the two modes of our multimode bosonic code. A second processor will be used for other modes.

Note that because the basic interaction converts two

Pumping gate:

$$\begin{aligned}
 |031\rangle &\mapsto |040\rangle, \\
 |011\rangle &\mapsto |020\rangle.
 \end{aligned}$$

This operation is necessary for reverting photon loss in the code mode stored in \hat{b} , while storing information about the occurrence of that loss in mode \hat{c} . Again, we will need two three-mode processors, each acting on one of the modes making up our error correcting code. Each of the physical modes of the code will be stored in the corresponding \hat{b} oscillators.

Entangling gate:

$$\begin{aligned} |011\rangle &\mapsto |011\rangle \\ |020\rangle &\mapsto \frac{|020\rangle + |002\rangle}{\sqrt{2}} \end{aligned}$$

This gate is also a symmetrizing operation for the state of the modes \hat{b} and \hat{c} . It is these two oscillators \hat{b} and \hat{c} that will contain the two modes of our error correcting code.

As we mentioned, the correspondence between the six modes of the two processors and the two modes of the bosonic code changes as the circuit runs. Initially figure 4 had a small inset describing this correspondence. We expanded on it in the supplementary materials.

FIG. 4. The error-correcting circuit, unrolled in time (the horizontal axis represents the flow of time, depicting how a pair of triply-resonant cavities is being used). This circuit would be executed on the hardware depicted in Fig. 3. The main drawing is the sequence of operations that we need to perform in parallel in the two triply-resonant cavities in order to perform the error correction. After placing the code and ancilla modes in the appropriate cavity modes, we accomplish the initial pumping and routing gates. After that, we need to shuffle the ancillary modes by releasing them in the appropriate waveguides. The spatial modes into which the code states are moved depend on the state of the ancillas, thanks to the conditional routing gates. As the ancillas contain information about the presence of photon-loss errors, this lets us perform operations conditioned on the loss of a photon, by performing the two conditional branches in parallel in different physical locations of the circuit. The conditional routing gates then act in reverse, ensuring that all spatial modes end in the same location, without breaking the bijectivity required for any quantum circuit. The various spatial modes employed can be seen in the bottom insets of the figure. The supplementary materials have a more detailed rendition. Importantly, as seen in Fig. 3, we do not need 12 triply-resonant cavities as depicted above, rather only 2 cavities with a network of waveguides and programmable beam splitters [44] that can route the spatial modes as necessary, so that each cavity can be used repeatedly.

Now the following figure is available in the supplementary materials:

The Two-mode Bosonic Code

$$|1\rangle_L = |22\rangle$$

$$|0\rangle_L = \frac{|40\rangle + |04\rangle}{\sqrt{2}}$$

Color-coding Legend

- One of the two modes of the bosonic code
- The other mode of the bosonic code
- Ancillary mode populated with one photon
- ⋯●⋯ Ancillary mode when unpopulated

(3) The measurement free QEC circuit is interesting. However, a similar idea and demonstration could be found in superconducting and trapped-ion systems. The authors should discuss previous works. Especially, the autonomous QEC was demonstrated recently.

We were certainly not intending to imply that all-unitary error-correction is a new concept. We have added references to previous schemes for all-unitary error correction. We have also added text discussing the differences between continuous dissipative schemes like AutoQEC and active all-unitary schemes such as ours.

quires quite a large number of ancillas and CNOT gates as discussed in the introduction. Our room-temperature design thus represents a dramatic reduction in circuit size and duration. We also emphasize that using all-unitary processes, which is the approach we take here, provides a practical advantage; doing so avoids the need to introduce additional amplification and classical feedback circuitry.

Competing with “active” gate-based approaches to measurement-free error correction, is the use of continuous autonomous QEC [55–59]. In that family of protocols one needs to design an exotic dissipator, usually through reservoir engineering, which provides an irreversible evolution from the error-space back the code space.

(4) In the model presented in Fig. 3, the photons are converted between different modes and also transferred between different cavities. It seems the authors forget to discuss the input-output of the cavities. For realizing high fidelity transfer between the cavities, there should be additional controllable coupling between the cavity and bus. It should be clarified.

We had briefly mentioned this in the section on “Examples of Programmable Gates”, but now we state it much earlier, during the initial description of the multimode cavity:

as for each subspace, there are many choices for $p(t)$ that generate the same unitary operation. We can use numerical search methods to find a control function $p(t)$ that simultaneously generates the required evolution for each of the set of subspaces. Naturally we wish to find the control that implements a given gate in the shortest time, a challenge solved as described below.

Lastly, the modes of this quantum processor will need to be actively coupled to the waveguides that carry the

quantum states to be processed. Otherwise, the process of capturing the content of the waveguides will be too slow due to the necessarily-high quality factor of the modes of the processor. To actively couple the cavity modes to the waveguides we envision using the method given in [23].

(5) The references about multiple mode bosonic codes should be added.

The following references discuss various single-mode and multi-mode bosonic codes.

- [25] M. Y. Niu, I. L. Chuang, and J. H. Shapiro, Qudit-basis universal quantum computation using χ (2) interactions, *Physical review letters* **120**, 160502 (2018).
- [26] M. Y. Niu, I. L. Chuang, and J. H. Shapiro, Hardware-efficient bosonic quantum error-correcting codes based on symmetry operators, *Physical Review A* **97**, 032323 (2018).
- [27] M. H. Michael, M. Silveri, R. T. Brierley, V. V. Albert, J. Salmilehto, L. Jiang, and S. M. Girvin, New class of quantum error-correcting codes for a bosonic mode, *Phys. Rev. X* **6**, 031006 (2016).
- [28] V. V. Albert, K. Noh, K. Duivenvoorden, D. J. Young, R. T. Brierley, P. Reinhold, C. Vuillot, L. Li, C. Shen, S. M. Girvin, B. M. Terhal, and L. Jiang, Performance and structure of single-mode bosonic codes, *Phys. Rev. A* **97**, 032346 (2018).

We added the following:

- [29] M. Bergmann and P. van Loock, Quantum error correction against photon loss using noon states, *Physical Review A* **94**, 012311 (2016).

(6) *It is not clear how to implement a gate between two logical qubits. The discussions should be extended.*

The following section (III.A.2) discussed that topic. We have now expanded it significantly. We felt the changes were too long to fit properly in this document, though, so we refer you to the manuscript.

2. Two-Qubit Logical Operations

Single-qubit rotations in the logical space of the two-mode code can be realized by using our three-mode processors, as such rotations do preserve $2n_a + n_b + n_c$. Moreover, two qubit logical operations can also be performed. For instance, consider a CPHASE gate, which together with the single-qubit rotations form a universal set. We need to perform the operation

$$|22\rangle \otimes |22\rangle \mapsto -|22\rangle \otimes |22\rangle,$$

(7) *For the hardware prospects, it is too long and distractive. The authors provide very detailed formula about the Hamiltonian, but they not directly related to the work presented in this work. To provide a more concrete discussion, other than hand-waving summarize of experimental progress, I would suggest the authors: First, provide the direct connection between the SHG efficiency and the N. Second, tell the guideline for experimentalists about how to optimize N. The Fig. 5&6 should be moved to supplementary materials unless the author could tell us which material is most promising and how to achieve the goal of realizing logical qubit.*

We moved parts of the derivation of the Hamiltonians to the supplementary materials. We kept some parts that we felt necessary to address comments from reviewer 2. The changes are too significant to present inline here. The link between SHG efficiency and N is now explicit and their forms are stated side by side.

(8) *A very important reference is missing: Optica Vol. 6, Issue 12, pp. 1455-1460 (2019) (a second-harmonic generation efficiency of 250,000%/W).*

Actually we did include this reference: it is reference 64 in the paragraph below. (It is also point number 20 in the scatter plot given in the supplementary materials.) The reference number has now changed in the new version of the manuscript. We also note that in wonderfully coincidental timing, a 20-fold improvement on this work was made public just this week. We have included a reference to this new development as well.

Considering some recent second harmonic generation (SHG) on-chip experiments (a $Q \sim 10^7$ in [62] and a $V_{\text{shg}} \sim 800\mu\text{m}^3 \sim 2000\frac{\lambda^3}{n^3}$ with a $70\mu\text{m}$ micro-ring in [62], at $\lambda_a \approx 750\text{nm}$) in a typical nonlinear optics

(9) The results in Fig. 7 is superficial. There are several ways to improve the discussion section of this work: (i) If the system has a finite Q, the state transfer between the cavities will also be imperfect. (ii) It is still not clear how to extend the code to more modes or more cavities. Is high order code possible? (iii) Could this architecture be generalized to fault-tolerant?

We have expanded the discussion of this topic. The architecture in principle can be extended to bosonic codes protected against more loss, but another practical option is to concatenate with a stabilizer code. It is almost certain that at some scale this concatenation will be necessary, as stabilizer codes with magic state distillation are one of the few concrete examples of scalable error corrected architectures. However, the particular choice of where to draw the line and switch from a small bosonic code to a concatenation with a stabilizer code is incredibly deep and application dependent. And this does not address the options of using another type of resource state (e.g. cluster states) at some intermediate level. Our goal is to provide a blueprint for the low-level physical system, that can accommodate many such architectural decisions.

Reviewer #2:

This manuscript is one of the most innovative and refreshing papers that I have read in the last two years. Krastanov et al. introduce a new photonic scheme for processing quantum information. The proposed advantages of this scheme are: 1) room-temperature operation; 2) realising sophisticated quantum-logic operations, such as the Toffoli gate or a conditional-routing gate, with low circuit area; and 3) likelihood of experimental realisation within the next decade. In both conception and explication this paper is a tour de force, and I expect it to have a sustained impact in quantum technology.

However there is one major claim that the paper makes repeatedly that I think the authors will have to modify. Statements such as:

“...may be the only feasible route to quantum information processing at room temperature” [Abstract] and,

“At present there appears to be only one platform with the potential for both room temperature and pressure operation: photonic circuits” [Introduction, 2nd para]

appear to be overclaims, as there appears to be other platforms with these characteristics.

Two examples are:

1. Metallic-like carbon nanospheres. Back in 2016 there was an observation of electron spin–lattice and spin–spin relaxation times of 175 ns at 300 K, published in Nature Communications (Náfrádi et al., vol. 7, 12232, 2016). This platform is now being developed commercially by Archer Materials, who in May this year signed a collaboration agreement with IBM.

2. Silicon–carbide defects. Last year there was a paper showing that an engineered quantum well can stabilize the charge state of a qubit, and compared this model to the experimentally observed room-temperature stability of the PL6 center in SiC. Again, this was published in Nature Communications (Iványi et al., vol. 10, 5607, 2019).

I had happened to read both these papers when they came out, but found them again quickly by doing a search for “room temperature quantum computer”. Of course there may be other platforms that I am unaware of, and I urge the authors to do a more comprehensive literature survey if they revise their manuscript.

We rectified the overly bullish language, and have acknowledged a number of alternative approaches. (Introduction, second paragraph, first sentence, and references 9-12)

With that out of the way, there are a few areas where the paper could be improved, and these arose, I think, because the authors know the material so well that they omit explanations or details that will be helpful even to expert readers. This manuscript should definitely be published, but I urge the authors to take the opportunity to clarify their presentation so as to maximise the impact of the paper.

p.1 Introduction. Second paragraph. When discussing room-temperature operation, there is no mention of the fact that currently the most efficient photon detectors are cryogenic; and that cryogenic detectors with good number resolution are slow (transition edge sensors) or that fast cryogenic detectors have very poor number resolution (superconducting nanowire single-photon detector, SNSPD). Omitting the problem of detection at this stage in the paper appears to be a major flaw in the argument for all room-temperature operation.

Indeed, even if all error-correction processes required for a computation are performed unitarily, measurements will need to be made at the end to produce classical output. At some level in a fault tolerant protocol it might also prove to be technologically useful to make measurements. We now make this point, and explain how our unitary circuits enable high-fidelity measurements to be made with inefficient detectors. (See our response to the next comment.)

p.2. Introduction. Fifth paragraph. Having made the claim that this manuscript is the only platform for room-temperature quantum information processing, the authors then

effectively side-step the still open question of detection: “However, we do not explore the topic of room temperature measurements at length in this work.”

I appreciate that such an exploration would add length to an already substantial paper (and indeed could be a lengthy paper in itself) but not addressing it in at least some depth in this manuscript substantially weakens the claim of all room-temperature operation. I think that only an expert will follow the logic in the single sentence: “...a unitary circuit can be used to map the Fock state of one photon to the Fock state of N photons” and even if they do so this leaves rather large open questions regarding efficiency, speed, and circuit area of such a scheme. It may be that all the advantages of this manuscript are negated by disadvantages of room-temperature detection. Of course, that may not be the case, but there is no way to tell from the arguments here.

We agree that it is worth explaining in a bit more detail how high-fidelity measurements can be made at room temperature using only inefficient detectors. The essential point is that under the same assumptions required for the operation of our unitary gates (that is, sufficiently strong nonlinearities combined with high-Q cavities) high-fidelity measurements could also be made using present-day room-temperature photon detectors. This can be achieved by a unitary circuit that transforms a single-photon state to a state with, say, 10 or more photons or by generating a phase shift in a state that already has a sufficient amplitude. There are many ways to implement the former. It can be done using a Chi-2 nonlinearity and set-up similar to our three-mode processor by implementing a doubling process: the Chi-2 converts 1 photon in a mode to two photons in a second mode using downconversion. Then, a Chi-2 is used with a classical pump to convert each of the two photons back to a photon in the first mode, leaving now two photons in the first mode. Repeating this cycle provides an exponential amplification with a linear number of operations. It doesn't represent a great deal of additional overhead if measurements are only used at the end of processing to read out a message or the result of a computation. We have expanded our discussion (the second-to-last paragraph of the introduction) to elucidate this.

Tangentially, as some additional proof of versatility, in the supplemental materials we now describe a protocol for efficient number-resolving detection that employs our hardware design together with an efficient non-resolving photodetector (the temperature requirements of that photodetector not being considered).

p.2. Methods. First paragraph. “We have set $\hbar = 1$ ”. Please do not do this. I know it is standard in theoretical quantum optics, but it is poor practice in a paper proposing a new experimental architecture, where eventually readers will wish to substitute in measured values to check the claims in the paper. (In the 90s there were several quantum optics proposals that could have been evaluated much more quickly had \hbar not been unity).

We agree that setting $\hbar=1$ can be confusing, even though we put back all the units when we talk about physical implementation later in the manuscript. We have now found an alternative to

setting $\hbar=1$, which both shows why \hbar does not need to feature when considering implementing the gates, and also shows explicitly how to convert back to full units. We believe this solves the problem.

Stefan: we should to copy the new text in here

have set $\hbar = 1$, and we are using the rate of the $\chi^{(2)}$ interaction (the term $\hat{a}\hat{b}^{\dagger 2}$) as our arbitrary unit of frequency. Time is thus measured using the inverse of that unit. We will not have to introduce any additional timescales until we consider loss in the section on hardware considerations. In that section we re-express these quantities in terms of the physics and engineering parameters of realistic devices.

p.3. II.B.1. “We can perform with high fidelity any gate that keeps $2n_a + n_b + n_c$ constant”. How high a fidelity? What limits it? In principle? In practice?

When ignoring dissipation and not constraining the length of the control pulses, the fidelities are arbitrary close to one. We now elaborate upon these constraints in the introductory paragraph of the section on programmable gates.

1. Examples of Programmable Gates

Through the use of our implicitly constrained optimal control method, we can perform with high fidelity any gate that keeps $2n_a + n_b + n_c$ constant. If the length of the control pulse is unconstrained, and dissipation is neglected, we can achieve fidelities arbitrary close to unity. For gates reported here, we constrain the duration of the control pulses as much as possible before reaching unitary fidelities lower than 0.999. In later sections considering hardware implementations, we also describe the effects of dissipation. Here we describe a number of important unitary operations that fulfill that constraint, some of

p.3. II.B.1.a. “We realise the gate in the Hadamard basis”. Where are the details of this scheme? I appreciate that you encode the three inputs of the Toffoli into the a, b, and c modes and apply your three-mode gate once, but what does $p(t)$ look like for this case? What is the fidelity, in principle? What are the effects of intra-cavity loss on the gate fidelity?

Our answer to this comment overlaps with the previous one. The fidelity can be arbitrary close to one if pulse duration is not constrained and dissipation is neglected. We provided $p(t)$ for a number of other more complicated gates, and given the random-like profiles of $p(t)$ derived

through optimal control, we elected to not include additional traces, as they do not contribute new understanding of the problem. However, all these traces are shown in the supplementary materials. More importantly, the optimal control code we are making public with this manuscript can be used to recreate pulses for any unitaries (preserving $2n_a+n_b+n_c$) within minutes (up to a few hours for high-photon-number gates). The in-depth discussion of how dissipation affects these fidelities is done in the hardware prospects sections. The tradeoffs between pulse duration and unitary fidelity is also briefly discussed in the supplementary materials (in the last two figures for example).

p.4. In the first 3 pages the expressions and equations are numbered, but this stops on this page. Please continue the numbering on pages 4–14.

All equations are now numbered.

p.4. Above the set of expressions that begins $|040\rangle \rightarrow |200\rangle$ it would be helpful if you reminded the reader of your encoding used at this point. It appears to be $|abc\rangle$, is this correct?

Now we have added a reminder of this notation.

p.5. II.B.2. Please provide details, perhaps as a section in the Supplementary materials, to validate the claims here that a single use of this architecture can also: 1. process unprotected single-photon states; 2. enable unitary modification of modes with high number of photons; and 3. perform number-resolved measurement on modes with high number of photons. (3. is particularly important given the earlier lack of detail on detection).

We added a new section (Additional examples of multiphoton operations) in the supplementary materials with a particular example of additional processing capabilities. The Toffoli gate and the simple CPHASE gate are already possible (in single or dual rail encoding) according to the main text, but we also introduce an interesting trick to turn multiple single photons into a single higher-photon-number state. Importantly, this unitary operation can be performed in reverse, leading to number-resolving detectors that do not need to rely on large trees of beam splitters. We clarified in the text that all of the “higher number of photons” operations we consider are still limited to small constants, only large enough to construct certain bosonic codes. We are not considering the asymptotic behavior where the number of photons is a scaling variable.

p.5. III.A. “We choose the two-mode code...” This would perhaps be clearer as: “We choose the two-spatial-mode code...”

We reworded the introductory sentence following this suggestion.

p.5 Above the set of expressions at the bottom of the right column, that begin $|1\rangle \otimes |C\rangle$, it would be helpful if you reminded the reader of your encoding used at this point. It appears to be $|c_1 c_2\rangle \otimes |b_1 b_2\rangle$, is this correct?

Yes, this is correct and we have added this reminder to the text.

pp.5–6. “The feedforward solution ... would be slow and require additional cooled hardware and classical decision circuitry”. Granted but won’t the small network shown in Figure 3 also require classical electronic control and possibly be slow? If the programmable beam splitters are implemented as Mach-Zehnder interferometers, as suggested, then the phase elements in these will have to be electronically controlled. To date such phase elements have most often be thermal, which is slow; electro-optic elements will be faster but also tend to introduce loss.

We expanded the figure description to specify that while classical electronics might be necessary to set the pacing of operations, no feedback or decision circuitry is necessary, because the protocol is measurement free. This significantly simplifies the engineering constraints for the device. Fast switches will indeed be required for this protocol to work. We briefly allude to the use of fast optical controlled switches. Lastly, we included a recent reference on experimental progress in this area (“Integrated lithium niobate electro-optic modulators operating at CMOS-compatible voltages”).

p.6 In the set of expressions at the bottom of the left column, it is not immediately obvious how the second and third unitary operations are realised. Could the authors please provide some details?

These operations preserve the $2n_a + n_b + n_c = \text{constant}$ of motion. This is the only constraint to our processors, and the optimal control algorithm easily finds a control pulse that performs the given operation. The process to do that is the same as for the operations already presented in fig 2. We will explicitly say this in the text and the code (in a readable and self-documenting format) that finds such pulses will be public.

p.7 Above the first set of expressions at the top of the left column, that begin $|00\rangle \rightarrow |22\rangle$, it would be helpful if you reminded the reader of your encoding used at this point. Is it $|b_1 b_2\rangle$?

Reviewer 1 had a similar suggestion. Recognizing that this section was too complex for the small amount of text dedicated to it, we have expanded it.

p.7 Above the second set of expressions at the top of the left column, that begin $|11\rangle \rightarrow |20\rangle$, it would be helpful if you reminded the reader of your encoding used at this point. Is it $|abc\rangle$ again?

A reminder to that effect is now added.

p.9. Hardware Prospects. Figure 7 caption and discussion in text. I'm afraid this confused me, as the Figure appears to show that encoded corrected logical qubit (green curve) at best matches the unprotected qubit (blue curve). Given the green curve never rises above the blue curve, how is the logical qubit beginning to outperform the physical qubit? This figure suggests the best thing to do is to just use the physical qubit, which I'm sure is not the intent of it.

In the text you say you "...see the error correcting threshold is $N \sim 2000$ ". How can one see this from this figure?

The plot was meant to represent the behavior of our system exactly at the break even point, similarly to the way this information is presented in seminal experimental work like <http://jianggroup.yale.edu/sites/default/files/nature18949.pdf>. At higher N we can perform more error correcting procedures per lifetime and the "error corrected" curve will be above the "no encoding curve". It is to be expected that between error correcting steps the fidelity of the encoded state will drop faster than the fidelity of the unencoded state, as there are more ways the decay channel can damage the state. But the larger states provide the redundancy necessary to reverse this process.

p.10. Last paragraph. "...our work ... shows that a single elementary photonic device can be reprogrammed on the fly..." Is there an analysis of reprogramming time in terms of device unit times, or is reprogramming assumed to be instantaneous?

There are a couple of ways we can approach this problem. If we have a number of pulse shapers, we can just use a simple switch that selects through which of them the control laser will pass. In this fashion reprogramming time is virtually instantaneous: the switching speed of the switch imposes the time between sending two different control pulses.

If we have only one wave shaper in the experiment, then the reprogramming time is set by how long it takes to set the device to a new pulse shape $p(t)$. This can be slow.

Lastly, deriving the pulses $p(t)$ is an operation that needs to be done in advance. A lookup table for such pulses can easily be implemented in high speed classical electronics, but preparing that lookup table would require tens of hours of optimization computations. We didn't feel it was necessary to include much of this discussion in the main text.

Reviewer #3:

The manuscript at hand theoretically describes a scheme for loss-protected optical quantum computing using bosonic codes.

The presented scheme is unique in that the error (loss) correction requires no measurements and feed-forward operations, but acts deterministically. This is enabled by presenting a novel 3-mode non-linear cavity architecture based on Xi(2).

The authors first present the elements of the scheme and then describe the error-correcting circuit as well as single- and two-qubit logical operations. They also provide a survey of the existing technology and conclude that the physical parameters required for the implementation of their scheme are achievable within the next decade.

The manuscript is very well written. It is clear, presents a novel, groundbreaking scheme, and projects its future feasibility by studying the existing technologies and their development in the past few years.

I think this is an extremely good manuscript, and I recommend its immediate publication without any changes.

Well done!

Reviewer #1 (Remarks to the Author):

The authors significantly revised their manuscript, and now I would recommend its publication in Nature Communication